# Effect of radiofrequency and pelvic floor muscle training in the treatment of women with vaginal laxity: A study protocol

**Gláucia Miranda Varella Pereira**[1], **Cássia Raquel Teatin Juliato**[1], **Cristiane Martins de Almeida**[2], **Kleber Cursino de Andrade**[2], **Júlia Ferreira Fante**[1], **Natália Martinho**[3,4], **Rodrigo Menezes Jales**[1], **Marcela Ponzio Pinto e Silva**[2], **Luiz Gustavo Oliveira Brito**[1]*

**1** Department of Obstetrics and Gynecology, School of Medical Sciences, University of Campinas, Campinas, Brazil, **2** Centro de Atenção Integral à Saúde da Mulher (CAISM)—Hospital da Mulher Professor Dr. José Aristodemo Pinotti—UNICAMP, Campinas, Brazil, **3** Centro Universitário das Faculdades Associadas de Ensino–UNIFAE, São João da Boa Vista, Brazil, **4** Centro Regional Universitário de Espírito Santo do Pinhal—UNIPINHAL, Santo do Pinhal, Brazil

\* lgobrito@unicamp.br

**Funding:** The first author GMVP receives Scholarship Grant from the São Paulo Research Foundation (FAPESP) - Grant 2019/26723-5.

## Abstract

### Background

Vaginal laxity is an underreported condition that negatively affects women's sexual function and their relationships. Evidence-based studies are needed to better understand this complaint and to discuss its treatment options. Thus, we present a study protocol to compare the effect of radiofrequency and pelvic floor muscle training in the treatment of women with complaints of vaginal laxity.

### Methods/Design

This is a prospective, parallel-group, two-arm, randomized clinical trial (Registry: RBR-2zdvfp–REBEC). Participants will be randomly assigned to one of the two groups of intervention (Radiofrequency or Pelvic Floor Muscle Training). The study will be performed in the Urogynecology outpatient clinic and in the physiotherapy outpatient clinic at the State University of Campinas–UNICAMP and will include women aged $\geq$ 18 years and with self-reported complaints of vaginal laxity. Participants will be assessed at baseline (pre-intervention period) and will be followed up in two periods: first follow-up (30 days after intervention) and second follow-up (six months after intervention).

### Expected results

The results of this randomized clinical trial will have a positive impact on the participants' quality of life, as well as add value to the development of treatment options for women with complaints of vaginal laxity.

### Trial registration

Registry: RBR-2zdvfp–Registro Brasileiro de Ensaios Clínicos–REBEC (19/02/2020).

**Competing interests:** The authors declare no competing Interests.

**Abbreviations:** IUGA, International Urogynecological Association; ICS, International Continence Society; PFMT, Pelvic Floor Muscle Training; RF, Radiofrequency; CAISM, State University of Campinas–UNICAMP; VLQ, Vaginal Laxity Questionnaire; GRA, Global Response Assessment; FSFI, Female Sexual Function Index; FSDS-R, Female Sexual Distress Scale-Revised; ICIQ-VS, International Consultation on Incontinence Questionnaire—Vaginal Symptoms; ICIQ-SF, International Consultation on Incontinence Questionnaire Short-Form; POP-Q, Pelvic Organ Prolapse Quantification; PFMS, Pelvic Floor Muscle Strength; PFMM, Pelvic Floor Muscle Morphometry; VT, Vaginal Thickness.

## Introduction

Vaginal laxity (VL) is defined by the International Urogynecological Association (IUGA) and the International Continence Society (ICS) as a complaint of excessive vaginal flaccidity [1]. This condition is rarely discussed between patients and their doctors, possibly due to the lack of evidence-based treatments, embarrassment and lack of knowledge in the assessment of this condition [2]. According to urogynecologists, VL still presents itself as an underreported condition with reports of discomfort that can affect sexual function and relationships [3, 4]. The way women perceive their genitalia has a strong and positive impact on their sexual function [5].

The prevalence of VL is 24% and appears to be associated with younger age, vaginal births, pelvic organ prolapse (POP) symptoms or physical exam findings. Therefore, it is a somatic, not psychogenic, dysfunction [6].

It is speculated that pregnancy and childbirth play a role in VL [3]. Although there is no proven link between VL and childbirth, research shows that vaginal delivery can result in pelvic floor injury [6, 7]. Pelvic floor and vagina trauma during pregnancy and vaginal delivery can lead to the lengthening of the vaginal opening leading to permanent changes in sexual and physical sensitivity during sexual intercourse. These changes promote an important reduction in the quality of life of women and their partnership [8, 9].

Potential consequences associated with vaginal delivery that extend beyond the postpartum period are urinary incontinence (UI), POP, chronic pelvic pain (CPP), and sexual dysfunction [10–13]. Not all women adapt to the psychological and physical changes in the postpartum period, which can lead to changes in the emotional relationship with their partner [14]. Both vaginal delivery and levator ani muscle trauma are associated with an increase in the diameter of the genital hiatus [15]. The genital hiatus is limited by the puborectalis muscle, a component of the levator ani muscle, and appears to play an important role in defining the high vaginal pressure zone [16]. Avulsion of the levator ani muscle, especially if proven bilaterally, would have some effect on female sexual function [17].

The diagnosis of VL has been based on patients' self-reporting. A comprehensive medical history, physical examination and psychosexual assessment are the initial steps to properly identify patients with VL [18].

The reduction in vaginal sensation during sexual intercourse may be related to anatomical damage to the perineal body, POP stage 1, laxity of the vaginal canal or introitus, damage to the nerves and connective tissue during pregnancy and childbirth or, potentially, a combination of these factors [19].

Surgical and non-surgical treatments for VL have been proposed. Surgical procedures for VL such as posterior colporrhaphy or perineorrhaphy are more commonly recommended. These procedures aim to reduce the size of the vaginal introitus, not necessarily treating the VL pathophysiology. Besides, 83% of the interviewed urogynecologists reported concerns with a potential risk for post-operative dyspareunia [3]. Post-surgical dyspareunia would further impair the quality of life of a woman who already complained of sexual dysfunction. Thus, it is necessary to develop non-surgical techniques that can assist in the treatment of other factors associated with VL, such as muscle hyperdistensibility and not just surgically reducing the size of the genital hiatus.

A non-surgical option for the treatment of VL includes pelvic floor muscle training (PFMT), which was initially recommended as a first-line treatment for UI [2, 20]. Pelvic floor muscle function appears to play an important role in female sexual function, and contraction of the levator ani muscle appears to increase the sexual response [21]. The contraction of the pelvic floor muscles also plays an important role in the female orgasmic response. Women

with weak muscles who receive pelvic floor rehabilitation and strengthen the muscles in that region perceive a positive effect on their sex life [22]. Pelvic floor muscle training could have an effect on the hypertensile muscles of women complaining of VL.

Another non-surgical therapeutic possibility to treat VL is radiofrequency (RF). Despite the scarcity of controlled clinical trials to evaluate the therapeutic advantages, safety and efficacy of radiofrequency [23], the studies carried out to date have shown good tolerance, as well as, subjective improvement of vaginal narrowing, sexual function and decreased sexual discomfort [2] with effects maintained by 12 months and without any adverse events [8]. RF seems to improve vaginal vascularization and collagen fiber reorganization, which may also contribute to a decrease in the sensation of VL [24].

To our knowledge, to date, no clinical trial has been developed to assess the role of pelvic floor muscle and radiofrequency training in VL. Thus, the general objective will be to compare the effect of RF and PFMT in women with VL symptoms. The specific objectives are related to the assessment of the sexual function, vaginal symptoms, and sexual distress, as well as, to assess the impact of UI on patients' quality of life. The POP staging, contractility, and pelvic floor muscle function will be also evaluated. Finally, we will assess the impression of improvement in VL complaints after the interventions.

Our hypothesis is that RF will be different from PFMT in treating women with VL symptoms.

## Materials and methods

### Trial design

This is a prospective, parallel-group, two-arm, randomized clinical trial. It involves three assessments in which primary and secondary outcomes will be evaluated: one pre-intervention visit, one 30-day post-intervention visit, and a six-month consultation after the intervention. Participants will be randomly assigned to one of the two groups of intervention (RF or PFMT). The study will follow the CONSORT recommendations [25] and the SPIRIT Statement (Standard Protocol Items: Recommendations for Interventional Trials) [26]. Fig 1 shows the detailed study steps.

The term VL was recently defined and little is known about this complaint. There is still no gold standard treatment for VL and further studies are needed to understand its pathophysiology. Although its pathophysiology is not completely known, there is a consensus on the association of VL with pregnancy and childbirth [2, 10, 27]. Some proposed mechanisms involve overstretching of the vaginal walls and introitus during vaginal birth and an increase in levator ani hiatal dimensions resulting from macro and microtrauma of the levator ani muscle [7, 17]. Although supervised PFMT is recommended as a first-line treatment for stress or mixed UI in women by most of the guidelines [28–30], more studies are needed to demonstrate the effect of PFMT on female sexual function. A randomized controlled trial concluded that women reporting improvement in sexual function demonstrated greatest increase in PFM strength and endurance [31].

Regarding RF, a recent randomized, multicenter, sham-controlled clinical trial found a statistically significant and clinically important improvement of VL with RF when compared with Sham treatment [32]. In our study, RF will be applied once every 4 weeks (a total of three applications) and will probably be less likely to face problems related to treatment adherence when compared to PFMT. Although the RF procedure has been shown to be well tolerated, adverse effects may occur [32, 33]. PFMT is generally free of adverse effects.

### Study setting

Patient recruitment and assessment/treatment will be carried out in the Urogynecology outpatient clinic at the School of Medical Sciences and in the Physiotherapy outpatient clinic at the

| | STUDY PERIOD | | | | | | | |
| --- | --- | --- | --- | --- | --- | --- | --- | --- |
| | Enrolment | *Baseline* | Post-allocation: Allocation, Interventions, Follow-up | | | | | |
| TIMEPOINT** | *February 2020 to June 2021* | 0 | Allocation | *1-4 w* | *5-8 w* | *9-12 w* | *F1* | *F2* |
| **ENROLMENT:** | | | | | | | | |
| Eligibility criteria | X | | | | | | | |
| Recruitment | X | | | | | | | |
| Initial Assessment | X | | | | | | | |
| Informed consent | X | | | | | | | |
| Allocation | | | X | | | | | |
| **INTERVENTIONS:** | | | | | | | | |
| *Radiofrequency* | | | | ◆——————◆ | | | | |
| *PFMT* | | | | ◆——————◆ | | | | |
| **ASSESSMENTS:** | | | | | | | | |
| *Sociodemographic Data* | | X | | | | | | |
| *Medical History* | | X | | | | | | |
| *BMI* | | X | | | | | | |
| *Sexual life Data* | | X | | | | | | |
| *Obstetric History* | | X | | | | | | |
| *Urinary/Intestinal Habits* | | X | | | | | | |
| *Physical Examination* | | X | | | | | X | X |
| *Female Sexual Function Index* | | X | | | | | X | X |
| *Female Sexual Distress Scale-Revised* | | X | | | | | X | X |
| *ICIQ-VS and ICIQ-SF* | | X | | | | | X | X |
| *Ultrasound Examination* | | X | | | | | X | X |
| *Global Response Assessment* | | | | | | | X | X |
| *Adverse Events* | | | | X | X | X | X | X |

**Fig 1. Description of the study steps.** F1: Follow-up (30 days after intervention); F2: Follow-up (6-months after intervention); w: week; PFMT: Pelvic Floor Muscle Training; BMI: Body Mass Index; ICIQ-VS: International Consultation on Incontinence Questionnaire—Vaginal Symptoms; ICIQ-SF: International Consultation on Incontinence Questionnaire Short-Form.

Centro de Atenção Integral à Saúde da Mulher (CAISM)—Hospital da Mulher Professor Dr. José Aristodemo Pinotti, both units affiliated to the State University of Campinas—UNICAMP.

## Study population

Women with self-reported complaints of VL. There is no objective and standardized diagnostic evaluation for VL and its pathophysiological mechanism is not yet known [34].

## Sample size

The sample calculation was based on the study by Krychman *et al.* [32], who demonstrated that RF therapy was associated with significant clinical and statistically significant improvement in sexual function in women with VL, when data analysis was performed in a group containing 73 patients. To calculate the sample of the present study, we used values of sexual function assessed using the FSFI questionnaire. There was an increase of 7 points in the FSFI score in the group treated with radiofrequency and an increase of 3 points in the control group. When considering a study power of 80%, an alpha of 0.05 with two-tailed test, it was found that the minimum number of participants required in each group will be added to a percentage of 30% loss in the sample, totaling 68 women, 34 in each group (isolated RF and isolated PFMT).

## Eligibility criteria

We will include women aged $\geq$ 18 and $\leq$ 60 years, with VL complaints assessed by direct question (yes / no) and by the VLQ [2] (very loose, moderately loose, slightly loose), and willing to attend treatments on the scheduled date and places.

Participants who present the following conditions will be excluded from the study: use of a pacemaker; decompensated heart disease; cognitive deficit; peripheral or central neurological disorders; the presence of any type of cancer; the presence of cervical dysplasia; history of active urinary or vaginal infection; decompensated metabolic diseases; patients undergoing physical therapy for pelvic floor disorders; patients using vaginal estrogen in the last 6 months; patients already undergoing surgery for prolapse or urinary or anal incontinence; patients with stage 2 POP onwards; force of contraction of the pelvic floor muscles equal to zero according to the modified Oxford scale [35].

## Recruitment

Women will be recruited from the Urogynecology outpatient clinic (CAISM / UNICAMP) and by social media advertisements, publicity posters, and printed ads from this study. All patients will be contacted by telephone by the researcher who applies the eligibility criteria and sends the VLQ via email or virtual communication platforms. The recruited participants will have their names, contact numbers, and VLQ responses recorded in a spreadsheet. A contact telephone with a virtual communication platform specific to this study is available for participants to contact the researchers whenever necessary. Participants will later be called by phone for the initial assessment. Participants interested in the study who did not pass the eligibility criteria are referred to the Urogynecology outpatient clinic for follow-up.

## Allocation

The randomization sequence will be carried out through a computer program, in a 1:1 allocation ratio, in two blocks. The numbers corresponding to the study groups (1. Radiofrequency

and 2. Pelvic Floor Muscle Training) will be placed in opaque sealed envelopes that will be opened by the study participants after signing the consent form and undergoing initial assessment in the first clinical visit.

The researchers who will assist in completing the questionnaires and physical examination, the researchers responsible for ultrasound and the data analysts will be blinded for the treatment group to which the participants were randomized to.

## Initial assessment (first clinical visit)

Patients registered on the recruitment spreadsheet will be contacted by phone and the initial evaluation will be scheduled. In the initial evaluation, the participants will go through a lecture given by the researcher (G.M.V.P) in order to present the study, the assessments, the interventions, and the follow-up periods. At this point, the participants will be able to have all questions answered about the study. Participants who agree to participate in the study will receive a consent form for reading and signing. Participants will have their personal details protected and a number will substitute their identities.

## Interventions

The intervention period for both groups will be 12 weeks.

**1- Radiofrequency.** The RF group will receive three radio frequency applications at 4-week intervals, (an initial application, a second application after four weeks, and a third application also after four weeks) comprising 12 weeks of intervention. The four-week period between applications will allow adequate healing of the vaginal tissues submitted to the application of radiofrequency. The procedure will be performed by a trained researcher (G.M.V.P) with a supervision of an experienced urogynecologist (C.R.T.J/L.G.O.B).

The Wavetronic 6000 Touch device with the Megapulse HF FRAXX system (Loktal Medical Electronics, São Paulo, Brazil) will be used, equipped with an electronic energy fractionation circuit, connected to a vaginal electrode with 64 microneedles 200μ in diameter and 1mm in length, and divided into an array of eight columns, with eight needles each. When pressing the trigger pedal, these 64 needles are not energized simultaneously and the energy release is randomized in columns of eight needles in a predefined sequence, which does not allow two adjacent columns to fire in sequence, preventing the thermal sum of the columns (control fractional firing system (*Smart Shoot*). This allows for cooling between the points and the preservation of tissues adjacent to the vaporized points, so that neocolagenesis and neoelastogenesis can occur, through fibroblastic stimulation [24].

*1.1-Procedure.* Topical anesthesia in the posterior vestibule and vaginal opening (mucosa) with 2% lidocaine gel, 2 to 3 minutes before the procedure (Table 1). A patient in a lithotomy position, with the lower limbs flexed and supported, will be introduced a disposable (high-impact polystyrene) vaginal speculum. A careful vaginal examination will be performed for any changes in the vaginal wall. Whiff test will be performed using a swab to collect vaginal discharge. A drop of 10% potassium hydroxide will be added over the vaginal secretion [36]. If a characteristic fishy odor is felt, the patient will not undergo RF and will be referred for evaluation of possible vaginosis. If the vaginal wall is intact and the whiff test is negative, the procedure will be continued, starting with topical anesthesia of the vaginal walls with 10% lidocaine spray. After 2 minutes, vaginal antisepsis with 2% aqueous chlorhexidine and cleaning with sterile 0.9% saline will be performed. After cleaning, the entire liquid content of the serum will be wiped with sterile gauze before starting to apply the RF.

The device will be calibrated in FRAXX mode, 45 Watts, Low (initial application) and Medium (second and third application) Energy program (40 and 60 milliseconds,

**Table 1. Pelvic floor muscle training and radiofrequency sessions according to the treatment duration.**

| Period | Interventions | | | |
|---|---|---|---|---|
| | **Radiofrequency** | **Pelvic Floor Muscle Training** | | |
| **1 to 4 weeks** | 1st application | 1st phase | 2nd phase | 3rd phase |
| | • 2% Lidocaine Gel<br>• Vaginal Examination with speculum<br>• Whiff Test<br>• 10% lidocaine spray<br>• Cleaning: 2% aqueous chlorhexidine and sterile 0.9% saline<br>• RF: 45Watts, Low Energy program (40 milliseconds)<br>• Post-procedure orientation: 10-day sexual abstinence | • PFM maximum contraction (6 r / sustained 6 s/ 6 rest: 1 time). Supine position.<br>• PFM maximal contraction + transverse contraction (6 r / sustained 6 s/ 6 s rest: 1 time). Supine position.<br>• PFM maximal contraction + hip elevation (6 r / sustained 6 s/ 6 s rest: 1 time). Supine position. | • PFM maximum contraction (1 cough / 3 r). Supine position.<br>• PFM maximal contraction with the lower limbs extended and abducted (6 r / sustained 6 s/ 6 s rest: 2 times). Supine position.<br>• PFM contraction in three stages—mild, moderate, maximum (6 r: 2 times). Sitting position.<br>• PFM maximal contraction (6 r / sustained 6 s/ 6 s rest: 2 times). Standing position. | • Pelvic mobilization (anterior and posterior tilts, lateral tilts and rotation of the pelvis). Standing position. No PFM contraction in this phase. 10 repetitions each pelvic movement. |
| **5 to 8 weeks** | 2nd application<br><br>Same procedure (above) | • PFM maximum contraction (6 r / sustained 8 s/ 8 s rest: 1 time). Supine position.<br>• PFM maximal contraction + transverse contraction (6 r / sustained 8 s/ 8 s rest: 1 time). Supine position.<br>• PFM maximal contraction + hip elevation (6 r / sustained 8 s/ 8 s rest: 1 time). Supine position. | • PFM maximum contraction (2 cough / 3 r). Supine position.<br>• Fast PFM maximal (8 r: 2 time). Supine position.<br>• PFM contraction in six stages—mild, moderate, maximum–maximum, moderate, mild (8 r: 2 times). Sitting position.<br>• PFM maximal contraction (8 r /sustained 8 s/ 8 s rest: 2 times). Standing position.<br>• PFM maximal contraction (8 r /sustained 8 s/ 8 s rest: 2 times). Four supports (hands and knees). | • Same Intervention (above) |
| **9 to 12 weeks** | 3rd application<br><br>Same procedure (above) | • PFM maximum contraction (6 r / sustained 10 s: 1 time). Supine position.<br>• PFM maximal contraction + transverse contraction (6 r / sustained 10 s/ 10 s rest: 1 time). Supine position.<br>• PFM maximal contraction + hip elevation (6 r / sustained 10 s/10 s rest: 1 time). Supine position. | • PFM maximum contraction (3 cough / 3 r). Supine position.<br>• Fast PFM maximal (10 r: 2 time). Sitting position.<br>• PFM contraction in six stages—mild, moderate, maximum–maximum, moderate, mild (10 r: 2 times). Sitting position.<br>• PFM maximal contraction (10 r /sustained 10 s/10 s rest: 2 times). Standing position.<br>• PFM maximal contraction (10 r /sustained 10 s/10 s rest: 2 times). Four to two supports (right hand and left knee/ left hand and right knee). | • Same Intervention (above) |

respectively), and the fractional vaginal electrode will be used. Applications will be under direct view. The electrode will be lightly pressed against the mucosa, without pressing, so that all microneedles are in uniform contact with the tissue. The application will be carried out sequentially on the lateral vaginal walls, in rows, avoiding overlapping, starting from the proximal third of the vagina to the distal third in the vestibule exposed by the opening of the speculum. The speculum will be gently rotated to the anteroposterior position for the application of the anterior and posterior vaginal walls. At the end of the procedure, the speculum will be gently removed. Patients will be advised on post-treatment care and the use of 5% dexpanthenol cream in the vaginal opening is recommended two to three times daily if there is any discomfort in the region for 2 to 3 days and interrupt sexual intercourse for 10 days. Patients will

be instructed to contact the researchers if they experience any discomfort or notice any changes in vaginal discharge. These patients will be evaluated by an experienced gynecologist. There will be three vaginal applications at 4-week intervals, totaling 12 weeks of treatment.

**2- Pelvic floor muscle training.** The patients in the PFMT group will be assisted during the sessions by an experienced physiotherapist (G.M.V.P). The participants of this group will have 12 individual sessions of supervised PFMT, lasting 40–60 minutes, once a week, and totaling 12 weeks of treatment and continue their treatment with home PFMT program. To follow the treatment at home, patients will receive a printed diary containing the complete PFMT program, with figures illustrating the positions and orientations for each exercise for the pelvic floor muscles. Patients will have the possibility to contact the physiotherapist for questions regarding home treatment by video call, audio call or messages through a telephone number made available exclusively for the study. Patients will be instructed not to perform other exercises for the pelvic floor, different from the exercises proposed by the intervention program of the present study. Our intervention program for PFMT is based on the studies of Bo *et al.* [37] and Dumoulin *et al.* [38].

*2.1- Procedure*. The first PFMT session is longer and focused on the careful evaluation of the pelvic floor muscles in order to identify any muscle condition that interferes with the progress of the intervention; guidance on the correct performance of the pelvic floor muscle contraction with the aid of vaginal palpation and educational material; presentation of the PFMT program; and finally, the first sequence of exercises.

Patients will be instructed to perform three phases of exercise in each session, with at least two sessions of the PFMT program per day (Table 1). The exercises will undergo progression every 4 weeks of intervention, totaling three progressions (6 repetitions, sustained for 6 seconds, 6 seconds for rest, 8 repetitions, sustained for 8 seconds, 8 seconds for rest, and 10 repetitions, sustained for 10 seconds, 10 seconds for rest). The first phase comprises three exercises of maximum contraction of the pelvic floor muscles in the supine position, with the lower limbs semiflected and the feet supported, associated with contraction transverse exercise, and hip elevation. The second phase comprises pre-contraction of the pelvic floor muscles associated with cough; maximum sustained contraction in the supine position with the lower limbs extended and abducted; fast contractions in both supine and sitting positions, contraction in three stages (mild, moderate, maximum) in a sitting posture; maximum sustained contraction in standing posture and four supports (hands and knees). The third phase comprises the relaxation period with breathing exercises associated with pelvic mobilization in a standing posture. No pelvic floor muscles contraction in this phase. Patients will be instructed on the importance of adhering to the PFMT. In case of two absences, the patients will be contacted and their permanence in the study will be reassessed.

## Primary outcomes

The primary outcome will be the indication of improvement in the VL symptoms after the proposed interventions assessed through a single question with seven possible answers. The Global Response Assessment (GRA) [2] is a 7-level scale with response to the question: "How are you now (levels of vaginal laxity/tightness and sexual satisfaction) compared to before treatment" (markedly improved, moderately improved, slightly improved, no change, slightly worse, moderately worse, markedly worse?). This scale item has been already used to evaluate improvement in vaginal laxity symptoms after treatment [2].

## Secondary outcomes

The description of secondary outcomes is described below together with their measurement instruments.

Female Sexual Function Index (FSFI) [39]: a brief and multidimensional instrument to assess sexual function in women. The questionnaire was developed and validated by Rosen *et al*. and consists of 19 items that investigate sexual response over the past four weeks and performance in six domains: sexual desire, arousal, lubrication, orgasm, satisfaction, and pain [39]. The validation in Portuguese took place in 2008 by Thiel *et al*. [40]. The answers are scored according to the sum of the items that make up each domain (simple score) and multiplied by the domain factor generating the weighted score [40]. The maximum score is 36 points, adding up the total of each domain. Wiegel *et al*. proposed a cut-off score to differentiate women with or without sexual dysfunction in the amount of 26.55 [41].

Female Sexual Distress Scale-Revised (FSDS-R): a self-report questionnaire with 13 questions scored on a 5-point Likert scale from 0 (never) to 4 (always) to assess the sexual distress. Sexual distress is characterized by a set of feelings (for example, unhappiness, guilt, frustration, stress, worry) and emotions that individuals have about their sexuality. It differs from sexual dysfunction related to symptoms of sexual function, such as arousal, orgasm and pain, separate from emotions [42]. The Portuguese translation was developed by Berenguer *et al*. [43].

International Consultation on Incontinence Questionnaire—Vaginal Symptoms (ICIQ-VS): is a 14-item questionnaire validated for the Portuguese language by Tamanini *et al*. that assesses the presence and the impact of vaginal symptoms, as well as their relationship with quality of life [44, 45].

International Consultation on Incontinence Questionnaire Short-Form (ICIQ-SF): validated in Portuguese by Tamanini *et al*., is a simple, brief, and self-administered questionnaire, capable of quickly and effectively assessing the impact of urinary incontinence on patients' quality of life. It consists of four questions that assess the frequency, severity, and impact of urinary incontinence. Your score can vary from 0 to 21, the greater the commitment, the higher the total value [46, 47].

Pelvic Organ Prolapse Quantification (POP-Q): the ICS recommends POP description and staging using this instrument [48, 49]. The staging classification is defined as [50] Stage 0: there is no demonstrated prolapse; Stage I: the most distal part of the prolapse is more than 1 cm above the level of the hymen; Stage II: the most distal portion of the prolapse is between 1 cm above the hymen and 1 cm below the hymen; Stage III: the most distal portion of the prolapse is more than 1 cm beyond the plane of the hymen, but everted at least 2 cm less than the total vaginal length; Stage IV: complete eversion or eversion of up to 2 cm from the total length of the lower genital tract. The hymen is the reference point used to describe the quantitative prolapse and represents the zero point. Six anatomical points will be evaluated with the aid of a disposable graduated ruler. Two on the anterior vaginal wall (Aa and Ba). Two on the posterior vaginal wall (Ap and Bp) and two points on the upper vagina (C and D). The genital hiatus, the total vaginal length and the perineal body will also be measured. All points will be measured in Valsalva maximum, except the total vaginal length [50]. The ICS clinically defined significant POP in stage II or higher [49, 50].

Pelvic Floor Muscle Strength (PFMS): the strength of the pelvic floor muscles will be graduated according to the modified Oxford scale (5-point) by means of bi-digital vaginal palpation with the patients in the supine position with the lower limbs supported [35].

Pelvic Floor Muscle Morphometry (PFMM) and Vaginal Thickness (VT): the morphometry of the pelvic floor muscles will be assessed during rest, during contraction of the pelvic floor muscles, and during the Valsalva maneuver [51]. The vaginal thickness will be assessed in its proximal, middle, and distal third using two approaches—transabdominal and transvaginal [52, 53]. The equipment used for transabdominal, transvaginal and transperineal ultrasound measurements will be the GE Voluson 730 Expert® (GE Medical System Kretz-Technik GmbH and Co OHG, Zipf, Austria). The 2 to 6 MHz convex RAB4-8L 3D / 4D probe will be

used to record the morphometry of the pelvic floor muscles. Measurements will be performed at rest, Valsalva maneuver, and pelvic floor muscle contraction with the patient in the supine position [51] for the angle of the levator plate, the anorectal angle, the thickness of the levator ani muscle, and the area of the levator hiatus in $cm^2$. For vaginal thickness, probes 4C-D 2 at 5 MHz transabdominal and 5 to 9 MHz transvaginal will be used to measure the vagina in its proximal, middle, and distal thirds [52, 53].

## Baseline assessment

After signing the consent form, patients will be submitted to an anamnesis that includes questions related to the date of birth, marital status, ethnicity, education, body mass index, physical activity, surgeries, prior diseases, and medication. Questions related to sexual life such as sexual activity (yes / no), type of sexual behavior (homosexual, heterosexual, bisexual), type of sexual intercourse (vaginal, anal or both), origin of VL complaint (participant, partner or both), time of VL symptoms (months). In addition, patients will be asked about their perception of VL symptoms. Questions related to obstetric history and urinary / intestinal habits will also be included.

Subsequently, patients who sign the consent form will be referred for a physical examination consisting of the assessment of the strength of the pelvic floor muscles and POP quantification. The questionnaires will be self-completed by the participants and collected at the end of the baseline assessment. Patients will then be referred for ultrasound exams. These procedures will be performed in the initial assessment (first visit).

## Follow-up period assessment

Patients will be followed up in two periods after the interventions: first follow-up (30 days after intervention) and second follow-up (six months after intervention). The assessment procedures in the follow-up period will be the same as those used in the baseline physical examination, questionnaires, and ultrasound exams. We will add the Global Response Assessment.

## Data collection and management

Researchers with over 20 years of research experience will coordinate data collection. An assistant researcher will be responsible for checking all signatures of the consent form and the answers to each questionnaire to ensure that there is no blank answer. A researcher will perform a physical examination of all patients in the study. An experienced physiotherapist (over 10 years) and specialist in women's health, under the supervision of the main researchers, will carry out both the interventions and different researcher with experience and expertise in ultrasound will perform the ultrasound exams. Patients will be contacted and monitored by telephone.

The analysis of the collected data will be preceded by the elaboration of a computerized database where the variables will be coded in a data dictionary and validated. This database will often be filled in by an assistant researcher and supervised by the main researcher.

## Harms

The suspension of the intervention will occur through the verification of significant levels of discomfort during the application of vaginal RF (Visual Analogue Scale), as well as the significant occurrence of events such as urinary tract infection, vulvovaginitis, irritation and vaginal injury after application of vaginal RF (telephone contact). In these cases, an appropriate

medical treatment will be offered. Women will be discontinued if they miss any radiofrequency sessions or if their presence in physiotherapy sessions does not reach 80%.

## Data analysis

Initially, a descriptive analysis of the data will be performed to characterize the research participants, in the form of values of absolute frequency and percentage (relative) for categorical variables and values of mean and standard deviation for numerical variables.

Statistical analysis of comparison and correlation of the obtained data will be performed. The Kolmogorov-Smirnov test will be performed to analyze the sample's normality. Depending on the results obtained in the normality test, the Analysis of Variance (data with normal distribution) or Wilcoxon and Mann-Whitney Test (non-parametric data) will be used for comparative analyzes between the groups. Likewise, Pearson or Spearman tests will be used for correlational analyzes. Categorical variables will be analyzed using the chi-square test or Fisher's exact test. Statistical analyzes will be performed using the statistical program SAS System for Windows (Statistical Analysis System), version 9.4, adopting a significance level of 5% (p <0.05). Study endpoints will be analyzed primarily for the per protocol population, and repeated, for sensitivity reasons, for intention-to-treat population [54].

## Ethical considerations

The present study has been analyzed and approved by the Research Ethics Committee of the State University of Campinas–UNICAMP–CAAE—12919119.9.0000.5404 (08/08/2019)–and CEP 3.495.558 (08/0/8/2019). This study is also registered in the Registro Brasileiro de Ensaios Clínicos–REBEC—RBR-2zdvfp as a clinical trial (19/02/2020).

All participants who agree to participate in the study will receive a consent form for reading and signing. In addition, participants will have their personal data protected and a number will replace their identities within the study.

## Trial status

This trial is currently recruiting participants for the study. The initial assessments have also been started. This study was initiated in 2019 and is planned to finish in 2023.

## Discussion

Vaginal laxity is a complaint that is still little discussed among patients and their physicians and the lack of evidence-based treatment negatively impacts the management of this condition. There is a need to evaluate non-surgical options that offer minimal adverse effects at lower costs for women with VL complaints.

The present study aims to investigate two types of non-surgical treatment for VL and to compare the effect of both therapies. If therapies prove to be equally effective for VL complaints, our study will open a path for non-surgical options for VL management. In addition, while we await evidence on the VL pathophysiology, our study will contribute to developing knowledge of treatment options for this condition that negatively affects women's sexual lives and relationships.

## Dissemination of study findings

The present study is a part of a Ph.D. thesis and its results will be presented to the scientific board of the State University of Campinas–UNICAMP and to national and international scientific conferences.

## Study amendments

Any protocol amendments that are necessary will be effectively communicated and modified in the relevant parties (trial registry, Research Ethics Committee, funding agency, and journal). Any inquiries regarding the study will be properly answered by the researchers in the initial assessment period and during the period of the study.

## Supporting information

**S1 Checklist. SPIRIT 2013 checklist study protocol.**
(DOC)

**S1 File.**
(DOCX)

**S2 File.**
(DOCX)

## Acknowledgments

The authors would like to thank the Hospital da Mulher Prof. Dr. José Aristodemo Pinotti—CAISM for authorizing data collection and interventions. We would like to thank the CAISM Ultrasound Service and its team for the partnership and assistance in our study, in particular to Dr. Renata Telles Piva Belluomini, Dr. Isabella Salvetti Valente and Dr. Maira Furtado Greco Mazzer. We would also like to thank the Physiotherapy Service for embracing our study by offering us its facilities and all support for our data collection and interventions. We also thank the Urogynecology outpatient clinics at the Faculty of Medical Sciences of the State University of Campinas, where our patients will be referred, and evaluated during the period of recruitment and data collection. Special thanks to the Department of Obstetrics and Gynecology at the State University of Campinas and the São Paulo Research Foundation -FAPESP.

## Author Contributions

**Conceptualization:** Gláucia Miranda Varella Pereira, Cássia Raquel Teatin Juliato, Luiz Gustavo Oliveira Brito.

**Data curation:** Gláucia Miranda Varella Pereira, Cristiane Martins de Almeida, Júlia Ferreira Fante, Rodrigo Menezes Jales, Marcela Ponzio Pinto e Silva, Luiz Gustavo Oliveira Brito.

**Formal analysis:** Gláucia Miranda Varella Pereira, Natália Martinho, Luiz Gustavo Oliveira Brito.

**Funding acquisition:** Gláucia Miranda Varella Pereira, Luiz Gustavo Oliveira Brito.

**Investigation:** Gláucia Miranda Varella Pereira, Cristiane Martins de Almeida, Júlia Ferreira Fante, Luiz Gustavo Oliveira Brito.

**Methodology:** Gláucia Miranda Varella Pereira, Cássia Raquel Teatin Juliato, Cristiane Martins de Almeida, Luiz Gustavo Oliveira Brito.

**Project administration:** Cássia Raquel Teatin Juliato, Luiz Gustavo Oliveira Brito.

**Resources:** Gláucia Miranda Varella Pereira, Cássia Raquel Teatin Juliato, Kleber Cursino de Andrade, Luiz Gustavo Oliveira Brito.

**Software:** Gláucia Miranda Varella Pereira, Cristiane Martins de Almeida, Kleber Cursino de Andrade, Natália Martinho, Luiz Gustavo Oliveira Brito.

**Supervision:** Cássia Raquel Teatin Juliato, Luiz Gustavo Oliveira Brito.

**Validation:** Gláucia Miranda Varella Pereira.

**Visualization:** Gláucia Miranda Varella Pereira, Cássia Raquel Teatin Juliato, Cristiane Martins de Almeida, Kleber Cursino de Andrade, Júlia Ferreira Fante, Natália Martinho, Rodrigo Menezes Jales, Marcela Ponzio Pinto e Silva, Luiz Gustavo Oliveira Brito.

**Writing – original draft:** Gláucia Miranda Varella Pereira, Cássia Raquel Teatin Juliato, Luiz Gustavo Oliveira Brito.

**Writing – review & editing:** Gláucia Miranda Varella Pereira, Cássia Raquel Teatin Juliato, Cristiane Martins de Almeida, Kleber Cursino de Andrade, Júlia Ferreira Fante, Natália Martinho, Rodrigo Menezes Jales, Marcela Ponzio Pinto e Silva, Luiz Gustavo Oliveira Brito.

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
