## [Decision Letter · Decision Letter 0]

24 Jun 2021

PONE-D-21-07963

EFFECT OF RADIOFREQUENCY AND PELVIC FLOOR MUSCLE TRAINING IN THE TREATMENT OF WOMEN WITH VAGINAL LAXITY: A STUDY PROTOCOL

PLOS ONE

Dear Dr. Brito,

Thank you for submitting your manuscript to PLOS ONE. After careful consideration, we feel that it has merit but does not fully meet PLOS ONE’s publication criteria as it currently stands. Therefore, we invite you to submit a revised version of the manuscript that addresses the points raised during the review process. You should address the reviewers comments, and also include in the abstract the outcome measures (at least the primary outcome). 

We look forward to receiving your revised manuscript.

Kind regards,

Jose María Blasco, Ph.D.

Academic Editor

PLOS ONE

Journal Requirements:

2. Please provide additional details regarding participant consent. In the ethics statement in the Methods and online submission information, please ensure that you have specified whether consent will be informed consent

3. Please include additional information regarding the surveys or questionnaires used in the study and ensure that you have provided sufficient details that others could replicate the analyses. For instance, if you developed the survey or questionnaire as part of this study and it is not under a copyright more restrictive than CC-BY, please include a copy, in both the original language and English, as Supporting Information. If the questionnaire is published, please provide a citation to the (1) questionnaire and/or (2) original publication associated with the questionnaire.

4. PLOS requires an ORCID iD for the corresponding author in Editorial Manager on papers submitted after December 6th, 2016. Please ensure that you have an ORCID iD and that it is validated in Editorial Manager. To do this, go to ‘Update my Information’ (in the upper left-hand corner of the main menu), and click on the Fetch/Validate link next to the ORCID field. This will take you to the ORCID site and allow you to create a new iD or authenticate a pre-existing iD in Editorial Manager. Please see the following video for instructions on linking an ORCID iD to your Editorial Manager account: https://www.youtube.com/watch?v=_xcclfuvtxQv

6. Please include your tables as part of your main manuscript and remove the individual files. Please note that supplementary tables (should remain/ be uploaded) as separate "supporting information" files

Additional Editor Comments (if provided):

Reviewers' comments:

Reviewer's Responses to Questions

**Comments to the Author**

1. Does the manuscript provide a valid rationale for the proposed study, with clearly identified and justified research questions?

Reviewer #1: Yes

Reviewer #2: Partly

2. Is the protocol technically sound and planned in a manner that will lead to a meaningful outcome and allow testing the stated hypotheses?

Reviewer #1: Yes

Reviewer #2: Partly

3. Is the methodology feasible and described in sufficient detail to allow the work to be replicable?

Reviewer #1: Yes

Reviewer #2: No

4. Have the authors described where all data underlying the findings will be made available when the study is complete?

Reviewer #1: No

Reviewer #2: No

5. Is the manuscript presented in an intelligible fashion and written in standard English?

Reviewer #1: Yes

Reviewer #2: Yes

6. Review Comments to the Author

You may also provide optional suggestions and comments to authors that they might find helpful in planning their study.

Reviewer #1: Line 52. Abstract � It is not recommended to use acronyms directly without having explained them beforehand and this makes the abstract difficult to read. In this case, the action that each of the two groups is going to carry out is not clear.

Line 52. Introduction � The data presented in this section are correct, however, it is necessary to restructure this section because the thread of the reading is lost, not making clear until the end what one wants to study. It is also necessary to delve a little into the benefits of the different non-surgical actions in this pathology, in order to correctly guide the introduction.

Line 119. Introduction � The secondary objectives are somewhat confusing, they seem like a simple list, please rewrite this point to make it clearer.

Line 131. Trial design � Why is an assessment not carried out immediately after the intervention? It would be interesting to see if there is any kind of change in that phase.

Line 171. Study population � Why not put a roof or old cut? Are the same results expected in the elderly as in the adult, for example?

Line 192. Eligibility criteria � Data already exposed are repeated with the study population section, please review it.

Line 301. Procedure � Is rest time allowed between exercise and exercise? If yes, how long is this break?

Line 326. Primary outcomes� Is the proposed scale validated? Some more explanation of the main variable of the study is missing, since it is not clear.

Line 326 to 402. Outcomes (primary and secondary) � It would be interesting to provide the interrater and test-retest reliability data for all the variables.

Line 419. Follow-up Period Assessment � indicate that the measurements are from the end of the intervention.

Line 430 � It would be convenient to name the years of experience that the researchers who carry out both the intervention and the data collection have.

Line 461 � Indicate version of the SPSS program

Line 449. Data analysis � This is considered a non-inferiority work, please indicate it and explain in this section how it will be verified.

Line 542. References � Please keep the same line spacing as the rest of the article.

Reviewer #2: A study protocol entitled „EFFECT OF RADIOFREQUENCY AND PELVIC FLOOR MUSCLE TRAINING IN THE TREATMENT OF WOMEN WITH VAGINAL LAXITY: A STUDY PROTOCOL” has interesting topic and currently scoping.

Some methodological inacurracies should be addressed before the paper can be considered for publication.

Major

1. Describe the rationale for using a non-inferiority design of the study.

2. What are study objectives? What kind of effects of radiofrequency and pelvic floor muscle training are expected by the authors?

3. Please justify the established efficiency of PFMT in Vaginal Laxity

4. Where is the first visit taking place? Is the assessor going to be blindfolded?

5. Line 210-222

Authors wrote: „The numbers corresponding to the study groups (1. Radiofrequency and 2. Pelvic Floor Muscle Training) will be placed in opaque sealed envelopes that will be opened by the study participants after signing the consent form and undergoing initial assessment or first clinical visit.”

Will the participants know the group they are allocated to after signing the consent form and undergoing initial assessment or first clinical visit ?

6. Line 223-225

Is the person performing physical examination of PFMbe going to be blindfolded or is it going to be the same person conducting Pelvic Floor Muscle Training ? Which centre are PFM trainings going to be conducted in?

7. Line 374

Will Pelvic Organ Prolapse Quantification (POP-Q) be the procedure to compare the efficacy of two treatment methods? Will Pelvic Organ Prolapse Quantification (POP-Q) be used as criteria of exclusion? In lines 196 - 202 the authors state: Participants who present the following conditions will be excluded from the study: ……patients with stage 2 POP onwards;…”

8. Please describe in detail how the vaginal wall thickness measurements will be defined and how they will be performed by transvaginal and transabdominal USI.

9. Line 393

Which method will be used for morphometry of PFM? Which ultrasound parameters will be assessed?

10. Are all USI examinations going to be performed by the same examiner ?

7. PLOS authors have the option to publish the peer review history of their article (what does this mean?). If published, this will include your full peer review and any attached files.

Reviewer #1: No

Reviewer #2: No

---

## [Author Response · Author response to Decision Letter 0]

23 Aug 2021

July 18th 2021

Dear Editors-in-Chief

Dr. Emily Chenette

PlosOne

I would like to thank you for the opportunity to have our study reviewed by the Plos One Editorial Board.

Dear Academic Editor

Dr. Jose María Blasco, Ph.D.

PlosOne

Thank you for considering our study for revision in this prestigious Journal. We believe that the comments will enrich enormously our work.

Our responses will be stated with track changes (highlighted in yellow) in the manuscript as recommended and in red in the present document.

Yours sincerely,

Luiz Gustavo Oliveira Brito MD/PhD

Corresponding Author (on behalf of the authors)

Journal Requirements:

Response: The style requirements were added as requested. Thank you.

2. Please provide additional details regarding participant consent. In the ethics statement in the Methods and online submission information, please ensure that you have specified whether consent will be informed consent

Response: We removed the information from Study Amendments and added the information in lines 485-487. Thank you.

3. Please include additional information regarding the surveys or questionnaires used in the study and ensure that you have provided sufficient details that others could replicate the analyses. For instance, if you developed the survey or questionnaire as part of this study and it is not under a copyright more restrictive than CC-BY, please include a copy, in both the original language and English, as Supporting Information. If the questionnaire is published, please provide a citation to the (1) questionnaire and/or (2) original publication associated with the questionnaire.

Response: The description of the questionnaires are found in lines 339-344 and 350-379. Thank you.

4. PLOS requires an ORCID iD for the corresponding author in Editorial Manager on papers submitted after December 6th, 2016. Please ensure that you have an ORCID iD and that it is validated in Editorial Manager. To do this, go to ‘Update my Information’ (in the upper left-hand corner of the main menu), and click on the Fetch/Validate link next to the ORCID field. This will take you to the ORCID site and allow you to create a new iD or authenticate a pre-existing iD in Editorial Manager. Please see the following video for instructions on linking an ORCID iD to your Editorial Manager account: https://www.youtube.com/watch?v=_xcclfuvtxQv

Response: The ORCID for the corresponding author has been provided. Thank you.

Responde: OK. Thank you.

6. Please include your tables as part of your main manuscript and remove the individual files. Please note that supplementary tables (should remain/ be uploaded) as separate "supporting information" files

Response: The tables have been included as required. Thank you.

Additional Editor Comments (if provided):

Reviewers' comments:

Reviewer's Responses to Questions

Comments to the Author

1. Does the manuscript provide a valid rationale for the proposed study, with clearly identified and justified research questions?

Reviewer #1: Yes

Reviewer #2: Partly

2. Is the protocol technically sound and planned in a manner that will lead to a meaningful outcome and allow testing the stated hypotheses?

Reviewer #1: Yes

Reviewer #2: Partly

3. Is the methodology feasible and described in sufficient detail to allow the work to be replicable?

Reviewer #1: Yes

Reviewer #2: No

4. Have the authors described where all data underlying the findings will be made available when the study is complete?

Reviewer #1: No

Reviewer #2: No

5. Is the manuscript presented in an intelligible fashion and written in standard English?

Reviewer #1: Yes

Reviewer #2: Yes

6. Review Comments to the Author

You may also provide optional suggestions and comments to authors that they might find 

helpful in planning their study.

Reviewer #1: Line 52. Abstract � It is not recommended to use acronyms directly without having explained them beforehand and this makes the abstract difficult to read. In this case, the action that each of the two groups is going to carry out is not clear.

Response: Thank you for your suggestion. Please find the correction in lines 50-51. 

Line 52. Introduction � The data presented in this section are correct, however, it is necessary to restructure this section because the thread of the reading is lost, not making clear until the end what one wants to study. It is also necessary to delve a little into the benefits of the different non-surgical actions in this pathology, in order to correctly guide the introduction.

Response: The Introduction has been changed as suggested. Thank you. Please find the text in lines 95-97; 102-111; 118-120, and 126-127.

Line 119. Introduction � The secondary objectives are somewhat confusing, they seem like a simple list, please rewrite this point to make it clearer.

Response: The secondary objectives have been corrected as required. Thank you. Please find the correction in lines: 130-134.

Line 131. Trial design � Why is an assessment not carried out immediately after the intervention? It would be interesting to see if there is any kind of change in that phase.

Response: We understand your question and appreciate your recommendations; however, it will not be possible to carry out the assessments immediately after the interventions due to the 10-day radiofrequency healing period. Therefore, due to healing and tissue repair, we decided to standardize the assessment after 30 days in both groups.

Line 171. Study population � Why not put a roof or old cut? Are the same results expected in the elderly as in the adult, for example?

Response: We thank you for this recommendation. Even though we are not unaware of the profile of the patient with vaginal laxity, we decided not to exceed 60 years of age. Please find this information in line 198.

Line 192. Eligibility criteria � Data already exposed are repeated with the study population section, please review it.

Response: We corrected the study population and inclusion criteria as suggested. Thank you. Lines 181-183, and 198.

Line 301. Procedure � Is rest time allowed between exercise and exercise? If yes, how long is this break?

Response: We included the rest time as required in the text and in table 1. Thank you. Lines: 316-317. 

Line 326. Primary outcomes� Is the proposed scale validated? Some more explanation of the main variable of the study is missing, since it is not clear.

Response: We added more information regarding the primary outcome as required. Thank you. Lines: 339-344.

Line 326 to 402. Outcomes (primary and secondary) � It would be interesting to provide the interrater and test-retest reliability data for all the variables.

Response: As we only have one investigator performing the interventions, we did not consider the interrater and test-retest reliability data.. 

Line 419. Follow-up Period Assessment � indicate that the measurements are from the end of the intervention.

Response: We have indicated the required information in line 428. Thank you.

Line 430 � It would be convenient to name the years of experience that the researchers who carry out both the intervention and the data collection have.

Response: We added the required information in lines 435-439. Thank you.

Line 461 � Indicate version of the SPSS program

Response: We updated the required information in lines 473-475.Thank you.

Line 449. Data analysis � This is considered a non-inferiority work, please indicate it and explain in this section how it will be verified.

Response: We thank you for raising this question, which made us reflect on the study design. Since our sample size calculation has not added the range of an inferiority margin, we decided to review the study design. We decided to carry out a comparison between the interventions. We made the modifications in the following lines: 48; 135-136; 140, and 145 (reference 25).

Line 542. References � Please keep the same line spacing as the rest of the article.

Response: We corrected the reference spacing as required.

Reviewer #2: A study protocol entitled „EFFECT OF RADIOFREQUENCY AND PELVIC FLOOR MUSCLE TRAINING IN THE TREATMENT OF WOMEN WITH VAGINAL LAXITY: A STUDY PROTOCOL” has interesting topic and currently scoping.

Some methodological inacurracies should be addressed before the paper can be considered for publication.

Major

1. Describe the rationale for using a non-inferiority design of the study.

Response: We thank you for raising this question, which made us reflect on the study design. Since our sample size calculation did not include an inferiority margin, we decided to review the study design. We decided to carry out a comparison between the interventions. We made the modifications in the following lines: 48; 135-136; 140, and 145 (reference 25).

2. What are study objectives? What kind of effects of radiofrequency and pelvic floor muscle training are expected by the authors?

Response: Please find this information in lines 118-120, 126-127, and 129-136.

3. Please justify the established efficiency of PFMT in Vaginal Laxity

Response: As we said in the lines 112-120 and 495-498, there are no stablished, gold-standard treatments for vaginal laxity yet. Thus, we decided to compare non-surgical treatment options for vaginal laxity.

4. Where is the first visit taking place? Is the assessor going to be blindfolded?

Response: We added information as required. Lines: 227-229; 232, and 428-429. The assessors will not know which group the participants will belong. 

5. Line 210-222

Authors wrote: „The numbers corresponding to the study groups (1. Radiofrequency and 2. Pelvic Floor Muscle Training) will be placed in opaque sealed envelopes that will be opened by the study participants after signing the consent form and undergoing initial assessment or first clinical visit.”

Will the participants know the group they are allocated to after signing the consent form and undergoing initial assessment or first clinical visit ?

Response: Study participants will only know which intervention group they will be allocated after they have signed the consent form and have undergone baseline assessment.

6. Line 223-225

Is the person performing physical examination of PFMbe going to be blindfolded or is it going to be the same person conducting Pelvic Floor Muscle Training ? Which centre are PFM trainings going to be conducted in?

Response: Thank you for raising this question. The person who will carry out the intervention will not be the same person who will carry out the assessment. Please find the correct information in lines 443-444.The training of the pelvic floor muscles will take place in the Urogynecology outpatient clinic at the School of Medical Sciences and in the Physiotherapy outpatient clinic at the Centro de Atenção Integral à Saúde da Mulher (CAISM) - Hospital da Mulher Professor Dr. José Aristodemo Pinotti.

7. Line 374

Will Pelvic Organ Prolapse Quantification (POP-Q) be the procedure to compare the efficacy of two treatment methods? Will Pelvic Organ Prolapse Quantification (POP-Q) be used as criteria of exclusion? In lines 196 - 202 the authors state: Participants who present the following conditions will be excluded from the study: ……patients with stage 2 POP onwards;…”

Response: Although we have excluded POP cases, the POP-Q classification is important because we would like to check if there will be changes in the measurements of genital hiatus, perineal body or other external measurement. 

8. Please describe in detail how the vaginal wall thickness measurements will be defined and how they will be performed by transvaginal and transabdominal USI.

Response: Please find the information required in lines 411-413

9. Line 393

Which method will be used for morphometry of PFM? Which ultrasound parameters will be assessed?

Response: Please find the information required in lines 408-411.

10. Are all USI examinations going to be performed by the same examiner?

Response: Yes and the examiner will not be aware of the study groups.

---

## [Decision Letter · Decision Letter 1]

25 Oct 2021

EFFECT OF RADIOFREQUENCY AND PELVIC FLOOR MUSCLE TRAINING IN THE TREATMENT OF WOMEN WITH VAGINAL LAXITY: A STUDY PROTOCOL

PONE-D-21-07963R1

Dear Dr. Brito,

We’re pleased to inform you that your manuscript has been judged scientifically suitable for publication and will be formally accepted for publication once it meets all outstanding technical requirements.

Kind regards,

Jose María Blasco, Ph.D.

Academic Editor

PLOS ONE

Additional Editor Comments (optional):

Reviewers' comments:

Reviewer's Responses to Questions

**Comments to the Author**

1. Does the manuscript provide a valid rationale for the proposed study, with clearly identified and justified research questions?

Reviewer #1: Yes

2. Is the protocol technically sound and planned in a manner that will lead to a meaningful outcome and allow testing the stated hypotheses?

Reviewer #1: Yes

3. Is the methodology feasible and described in sufficient detail to allow the work to be replicable?

Reviewer #1: Yes

4. Have the authors described where all data underlying the findings will be made available when the study is complete?

Reviewer #1: Yes

5. Is the manuscript presented in an intelligible fashion and written in standard English?

Reviewer #1: Yes

6. Review Comments to the Author

You may also provide optional suggestions and comments to authors that they might find helpful in planning their study.

Reviewer #1: Dear authors,

After reviewing the corrections, it has been observed that all suggested changes have been noted to have been corrected or a rational response to the proposed suggestion justified.

Both the design and the language of the article are considered to be more than correct.

In addition, the article is considered to meet the minimum standards of the journal for publication.

finally indicate that for my part it is not necessary to make any additional changes.

Sincerely

7. PLOS authors have the option to publish the peer review history of their article (what does this mean?). If published, this will include your full peer review and any attached files.

Reviewer #1: No

---

## [Editor Report · Acceptance letter]

29 Oct 2021

PONE-D-21-07963R1 

Effect of radiofrequency and pelvic floor muscle training in the treatment of women with vaginal laxity: a study protocol 

Dear Dr. Brito:

I'm pleased to inform you that your manuscript has been deemed suitable for publication in PLOS ONE. Congratulations! Your manuscript is now with our production department. 

Kind regards, 

on behalf of

Dr. Jose María Blasco 

Academic Editor

PLOS ONE